# Bacterial Co-Infection in Patients with Coronavirus: A Rapid Review to Support COVID-19 Antimicrobial Prescription

**María Fernanda Celaya Corella, Jorge Omer Rodeles Nieblas** , **Donato Antonio Rechy Iruretagoyena**
and **Gerson Ney Hernández Acevedo \***

Department Clinical Microbiology and Parasitology, Faculty of Medicine Mexicali, Autonomous University of
Baja California, Baja California 21000, Mexico; maria.fernanda.celaya.corella@uabc.edu.mx (M.F.C.C.);
rodeles.jorge@uabc.edu.mx (J.O.R.N.); rechy@uabc.edu.mx (D.A.R.I.)
* Correspondence: ney.hernandez@uabc.edu.mx; Tel.: +52-686-557-1622

**Abstract:** The goal of this study was to determine the most common microorganisms present in COVID-19 patients with co-infections at the General Hospital of Mexicali. Bacterial co-infections have been reported in two previous global pandemics caused by viruses. In a retrospective observational study, we obtained information from 1979 patients. All had symptoms of respiratory disease, and we performed real-time Polymerase Chain Reaction tests on nasopharyngeal swab samples. Of the 1979 patients, 316 were negative; so, they were excluded. One thousand and sixty-three patients had positive results for COVID-19; one hundred and seventy-two (10.34%) had respiratory co-infections. These data were corroborated by positive growth results in culture media and identified using the MALDI-TOF MS System. Vitek 2® Compact, an automated identification system, determined the antimicrobial susceptibility testing results. We analyzed and determined the microorganisms in co-infected patients. Different microorganisms were found, including bacteria and fungi. The most prevalent of all the organisms was *Acinetobacter baumannii*, which was present in 64 patients (37.2%). We recommend improving the diagnostic and surveillance protocols for possible cases of co-infections in patients with COVID-19. Unlike co-infections in pandemic influenza, the spectrum of microorganisms that cause COVID-19 is too broad and varied to recommend empiric antibiotic therapy.

**Keywords:** co-infection; SARS-CoV-2; antimicrobial resistance; empiric antibiotic stewardship



## 1. Introduction

The first case of Coronavirus Disease 2019 (COVID-19) was reported in December 2019, and by 14 January 2022, a total of 318 million confirmed cases had been reported, with 5.5 million confirmed deaths [1]. Since its first detection, the infection and mortality rates of Severe Acute Respiratory Syndrome Coronavirus 2 (SARS-CoV-2) have far exceeded those of any other common flu. The World Health Organization declared the novel coronavirus outbreak a public health emergency of international concern. The co-infection of SARS-CoV-2 with other microorganisms, such as other viruses, bacteria, and fungi, is possible in COVID-19 for: The possibility of co-infection of SARS-CoV-2 with other microorganisms, including viruses, bacteria, and fungi in COVID-19, is present [2,3]. It can raise the difficulties in the diagnosis, treatment, and prognosis of COVID-19 and even worsen the disease symptoms and mortality risk [4]. There is a clinical need for the robust investigation of co-infections among patients with COVID-19. Bacterial co-infections have been observed during previous viral pandemics, including the 1918 influenza pandemic and the 2009 influenza A (H1N1) pandemic, with *Streptococcus pneumoniae (S. pneumoniae)*, *Haemophilus influenzae (H. influenzae)*, and *Staphylococcus aureus (S. aureus)* being the most common causative pathogens of respiratory tract infections [5]. Bacterial co-infections in COVID-19 have been a widespread concern amongst healthcare professionals due to overlapping clinical features with bacterial pneumonia and the increased risks of morbidity and mortality

associated with bacterial co-infection for: Bacterial co-infections in COVID-19 have been a widespread concern amongst healthcare professionals due to overlapping clinical features with bacterial pneumonia and the increased risks of morbidity and mortality [6].

Many studies on hospitalized patients with COVID-19 note the empirical use of antibiotics in most patients. Some clinical guidelines have recommended empirical antibiotic therapy to treat suspected bacterial respiratory co-infections in COVID-19 patients [7]. Therefore, tools to support and promote antibiotic stewardship in this population are needed [8]. Antibiotics are often empirically prescribed to COVID-19 patients, as they are in most illnesses. Distinguishing between viral pneumonia and a bacterial co-infection at presentation and during infection with COVID-19 disease can be challenging due to various similarities, including their characteristically high inflammatory markers and the frequent presence of pulmonary infiltrates on chest X-ray or computed tomography imaging [8]. Only 28% of patients with COVID-19 had enough sputum to develop a Gram stain, even though 98% of patients exhibited bilateral lung involvement on chest X-rays [9].

However, for healthcare-associated or secondary infections, like invasive mechanical ventilation, short-term peripheral venous catheters may act as risk factors for commonly observed complications in COVID-19 patients, such as sepsis and ventilator-associated pneumonia. In patients with a clinically suspected bacterial infection or pneumonia, empirical antimicrobial therapy is recommended to be quickly initiated even if a causative microorganism cannot be unequivocally identified and often before the results of microbiological diagnostics are available. Antimicrobial treatment might prevent secondary infections and reduce the complication rates [10]. In the context of rising levels of antimicrobial resistance, this study aims to inform sustainable and judicious antibiotic use. Therefore, there is concern about the overuse of antibiotics in managing COVID-19, with the attendant risk of an increase in the prevalence of antimicrobial resistance in the affected populations. In Latin America, the empirical use of antibiotics in the context of the pandemic has been documented and worsens the potential for antimicrobial resistance to available drugs. Enterobacterales harboring multiple carbapenemase genes likely resulted from the same or an increased trend of high antibiotic use, particularly the most potent $\beta$-lactams, which have been reported in pre-pandemic years.

The pandemic might have accelerated the problem of resistance because of antimicrobial drug misuse, prolonged hospital stays, and the high rate of antimicrobial prescriptions in hospital settings, such as medical wards and ICU, where many patients with COVID-19 were hospitalized during the first pandemic wave. Updated regional data about the use of antimicrobial drugs during the pandemic are required to confirm these assumptions. For instance, more than half of the responding countries in a World Health Organization (WHO) survey reported increases in the total number of prescriptions of antibiotics during the COVID-19 pandemic [11]. Several bacterial pathogens isolated from COVID-19 patients display resistance to multiple antibiotic classes. This pandemic has placing a strain on the resources of healthcare systems worldwide. The evidence presented in this article can inform better treatment and the more efficient use of equipment, medication, and time [12].

## 2. Materials and Methods

In this article, we used data obtained over approximately eleven months from 14 December 2020 to 10 November 2021; with the authorization of Hospital General de Mexicali, we obtained basic information from 1979 patients. All hospitalized patients presented symptoms of respiratory disease or had elevated risk factors (ex., close contact with a known infected patient). This necessitated the realization of a Real-Time Polymerase Chain Reaction test (RT-PCR test) of nasopharyngeal exudate. The positive growth results in culture media were corroborated and identified through Matrix-assisted laser desorption/ionization time-of-flight mass spectrometry (MALDI-TOF MS; MALDI Biotyper® System. Bruker Daltonics, Billerica, MA, USA), under the control of FlexControl software (version 3.0; Bruker Daltonics). We performed antimicrobial susceptibility testing using a Vitek 2® Compact (Biomérieux, Inc., Hazelwood, MO, USA) identification system machine.

The Vitek 2® Compact is highly automated and uses very compact plastic reagent cards (credit card size) that contain microliter quantities of antibiotics and test media in a 64-well format. The Vitek 2® Compact employs the repetitive turbidimetric monitoring of bacterial growth during an abbreviated incubation period [13].

The whole process of identifying the microorganisms and getting an antibiotic resistance profile takes less than twelve hours. Since the entire process is automated and requires only a little human input, the margin of error is minimal. Patients' data were obtained through a local census from different months used by other personnel in different areas and services of the hospital. We then organized these data on an individual Google Sheets drive file, where the patients were sorted by positive and negative results of the RT-PCR test. Then, via email, we searched where the laboratory info was sent for all 1663 patients to see if they had culture media growth results; 172 had positive growth on culture media, and culture media with no development was not considered for this article. Cultures taken from a place other than airway secretions were also not considered. All 172 patients were then organized on a different Google Sheets drive archive, where they were arranged according to which microorganisms had grown. Other patient data, like age or gender, were not considered since not every census included these data.

## 3. Results

Of the 1979 patients, 316 resulted in a negative result, so they were excluded from this article. Of the 1663 patients who tested positive for the RT-PCR test, 172 (10.34%) had positive results in the culture media. Different microorganisms were found, including bacteria and fungi. The most prevalent of all organisms was *Acinetobacter baumannii (A. baumannii)*, which was present in 64 patients (37.2%). The second most common microorganism was *Pseudomonas aeruginosa (P. aeruginosa)*, which was present in 29 of the 172 patients (16.9%). The third and fourth most common microorganisms were *Klebsiella pneumoniae* and *Stenotrophomonas maltophilia (S. maltophilia)*, which were present in 24 patients each (14.0%). The most common fungi were *Candida* sp. (*glabrata, tropicalis, albicans*, and *dubliniensis*), which were present in 11 of the 172 patients (6.3%), and a total of 35 different microorganisms were identified (Figure 1).

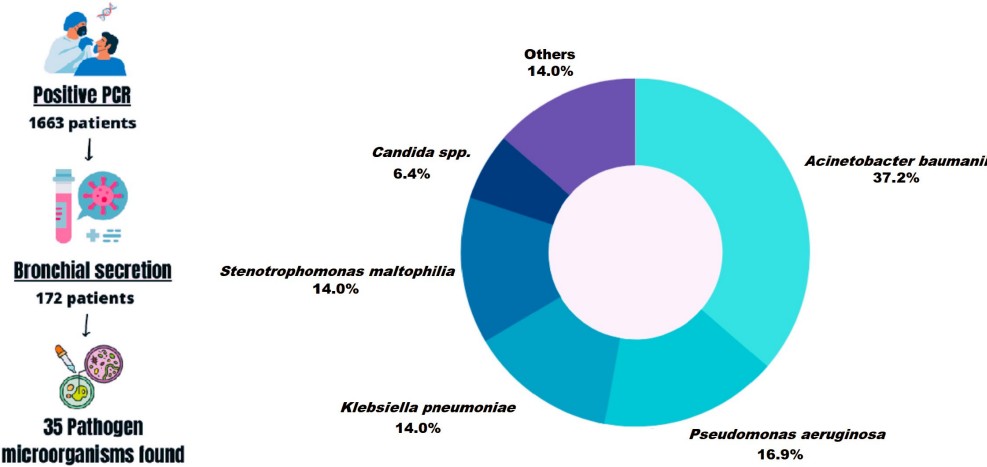

**Figure 1.** General scheme of methodology and identification of the co-infections.

We took the four most prevalent microorganisms in co-infections as a reference. Of the 64 patients co-infected with *Acinetobacter baumannii*, 41 patients (64%) died. Of the 29 patients that had a co-infection with *Pseudomonas aeruginosa*, 22 of them (75.8%) passed away; out of the 24 patients co-infected with *Klebsiella pneumoniae*, 18 of them (75%) died, and out of the 24 patients co-infected with *Stenotrophomonas maltophilia*, 17 of them (70.8%) passed away. All these patients died while hospitalized due to complications related to their co-infection (Figure 2).

## Confirmed co-infected cases and deaths

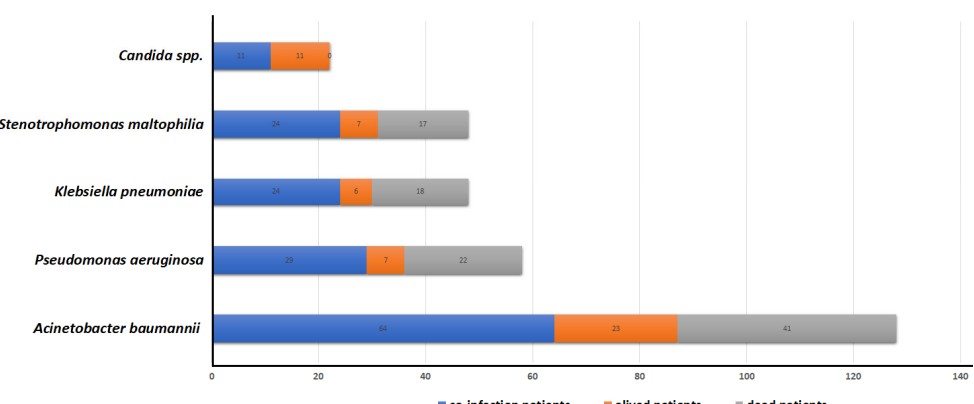

**Figure 2.** Of the 64 patients co-infected with *Acinetobacter baumannii*, 41 (64%) passed away, 29 had co-infection with *Pseudomonas aeruginosa*, and 22 (75.8%) died. Eighteen of twenty-four patients co-infected with *Klebsiella pneumoniae* died, and out of the twenty-four co-infected with *Stenotrophomonas maltophilia*, seventeen of them (70.8%) passed away. All the patients died while hospitalized due to complications related to their co-infection.

## 4. Discussion

Over 323 million cases of COVID-19 were confirmed as of January 2022 globally, and over 5.5 million deaths were reported at that point [14], representing a lethality of 1.7%. There was a slowdown in the increase in the case incidence and lethality data. By that time, multiple vaccines were already available and had been globally distributed. These previous data refer to general overall data. If we look for particular data, we can see that certain patients' mortality rates were way higher. This article focuses on hospitalized patients with a confirmed SARS-CoV-2 infection and a confirmed respiratory co-infection. We compare global data from systematic reviews and meta-analyses to local data from the General Hospital of Mexicali.

The first meta-analysis [15,16] indicated that 7% of the hospitalized patients had a bacterial co-infection, which increased to 14% in the studies focused on ICU (Intensive Care Unit) patients. In this meta-analysis, the most commonly detected pathogen was *Mycoplasma pneumoniae*, followed by *Pseudomonas aeruginosa, Haemophilus influenzae,* and *Klebsiella pneumoniae*. This meta-analysis does not mention the overall mortality of these patients since it was widely different according to the hospital and patient situation, for example, if the patient was in the ICU. This difference is evident if we compare the data with the results obtained locally through our investigation. We considered 1663 patients hospitalized with a confirmed COVID-19 infection through PCR. Of those 1663 patients, 172 (10.3%) had a confirmed bacterial or fungal co-infection. The most commonly found pathogen in our data was *Acinetobacter baumannii*, which is only mentioned as uncommon, but is a deadly cause of co-infections [16] in some hospitals. Other pathogens were also found, but their numbers vary widely when they are compared. *Mycoplasma pneumoniae*, the most common pathogen in the first meta-analysis, was not even found in a single patient as a cause of respiratory co-infections in our local data. These data are also largely different if we compare the most common pathogens found in influenza co-infections, which were very consistently *Staphylococcus aureus* and in some cases *Streptococcus pneumoniae*, which are well-studied phenomena, the same cannot be said about co-infections with COVID-19, as basically, the pathogens causing co-infections are broadly variable for The data on co-infections during influenza show that *Staphylococcus aureus* and, sometimes, *Streptococcus pneumoniae* are the most common pathogens. These findings are well-established. However, co-infections with COVID-19 have a wide range of pathogens involved, making it difficult to draw conclusions [17].

The previous data are relevant as empiric antimicrobial therapy was broadly used during the pandemic for The previous data is relevant because empirical antimicrobial therapy was widely used during the pandemic. Some articles [15,18] mention the prescription of early empiric antimicrobial treatments to up to 56% of all hospitalized patients with COVID-19 for: Some articles [15,18] mention the prescription of early empiric antimicrobial treatments to up to 56% of all hospitalized patients with COVID-19; in that particular article, the most commonly used therapies were Ceftriaxone (38%), Vancomycin (13%), Doxycycline (10%), and Cefepime (10%); in the same article, bacterial co-infections were only proven to be present in 3.5% of all patients, which means that the grand majority of patients received unnecessary antimicrobial therapy [19,20]; the authors mentioned that 52/147 (35%) patients received empirical antibiotics and that 19 (37%) received antibiotics for more than a week despite having negative cultures. The median length of the course of empirical antibiotics was seven (IQR = from 5 to 12) days. Rothe and colleagues [10] described implementing an antibiotic stewardship standard operational procedure in their institution in which initiating antibiotic therapy was recommended only in cases of clinically suspected infections (narrow spectrum Aminopenicillin/beta-lactamase inhibitor combination). However, stewardship decisions were made at the clinician's discretion. The medical reasoning supporting this decision for COVID-19 patients is to treat possible community-acquired bacterial pneumonia and the difficulty in distinguishing between bacteria- and virus-related symptoms, given that both cause unspecific symptoms such as coughing and a fever [21]. Besides the risk related to the secondary effects of different medications, another increasingly worrying problem with the broad use of empiric antimicrobial therapy is the rise of antimicrobial resistance. Multiple reports have proven that the incidence of resistant bacteria has increased, with statistics as big as a rise from 6.7% in 2019 (pre-pandemic) to 50% in April 2020 of Carbapenem-resistant Enterobacteriaceae [19]. We can also see the incidence of found resistance at the National Taiwan University Hospital. Most antibiotic agents increased their resistance rate between January to June of 2019 and one year later in the same period [19]. Overusing antimicrobials increases the risk of multi-resistant nosocomial secondary infections associated with unfavorable clinical outcomes. Therefore, empirical antibiotic coverage in COVID-19 patients must be carefully evaluated [10]. We recommend implementing rapid-result diagnostic techniques such as the BIOFIRE® FILMARRAY® Pneumonia Panel Plus (BIOMÉRIEUX) in another pandemic, allowing the early diagnosis of co-infections [22].

Close attention should be paid to those patients with SARS-CoV-2 who are admitted to the ICU. They present high risks of a severe disease, even a risk of death. Studies of the main factors in patients with a poor prognosis were conducted from the very initial moment that these patients were admitted. Age, blood glucose, cardiac and renal function, diabetes, and the inflammatory state were the main factors analyzed. The SARC-CoV-2 virus penetrates lung cells through endocytosis mediated by the Angiotensin 2 receptor, increasing its viral load through viral replication. More of the virus is released by budding or by inducing programmed cell death. After recognition by pattern recognition receptors in the body's immune cells, many cytokines are released via signal transduction, activating more immune cells to clear the virus, forming a storm of cytokines. An overactivated immune system will certainly kill a large number of lung cells. This damage will lead to respiratory failure and, eventually, death from hypoxia. Artificial ventilation is essential for these patients and is a very relevant procedure in bacterial co-infections. This risk increases when the patient remains intubated for extended periods [23]. We suggest further research on optimal ventilation strategies for patients with COVID-19 in the ICU. The regional differences in the results imply the need to develop specific regional protocols for ventilatory support and general treatment [24]. Studies of the resistance profiles of the most prevalent bacteria in the region and the ability to adjust them according to local needs are relevant [25]. Regional differences also imply the need for region-specific and flexible treatment protocols, especially in resource-limited settings. Instead of establishing general objectives and uniform international protocols for treating COVID-19 patients admitted to

ICUs and mechanical ventilation, we recommend adjusting them to particular communities. As COVID-19 continues to cause a critical problem, we need to quickly investigate the effects of bacterial co-infections during viral infections and investigate new antibiotics to eradicate multi-resistant pathogens. Furthermore, it is possible to combat the spread of multidrug-resistant bacteria with preventive measures [26].

## 5. Conclusions

The general use of antimicrobial therapy during the pandemic, especially in the beginning, definitely had enormous repercussions on the rate of antimicrobial resistance around the globe. The consequences of using antimicrobial medication have already started to show. Based on previously mentioned data, we recommend improved diagnostic protocols and vigilance regarding possible cases of co-infections among patients with COVID-19. Unlike co-infections in the influenza pandemic, the spectrum of microorganisms causing co-infections with SARS-CoV-2 is too extensive and varied to recommend empiric antibiotic therapy. A multidisciplinary approach is essential for optimal patient care via diagnostic stewardship and antibiotic programs because mortality is reduced when the correct antimicrobial treatment based on knowledge of local resistance is promptly initiated." For: A multidisciplinary approach is crucial for optimal patient care, including diagnostic stewardship and antibiotic programs. Mortality rates decrease when prompt initiation of the correct antimicrobial treatment based on local resistance knowledge occurs.

**Author Contributions:** Conceptualization, M.F.C.C. and J.O.R.N.; methodology, M.F.C.C., J.O.R.N., D.A.R.I. and G.N.H.A.; validation, D.A.R.I. and G.N.H.A.; formal analysis, M.F.C.C., J.O.R.N., D.A.R.I. and G.N.H.A.; investigation, M.F.C.C. and R.J; resources, M.F.C.C. and R.J; data curation, M.F.C.C. and J.O.R.N.; writing—original draft preparation, M.F.C.C., J.O.R.N. and G.N.H.A.; writing—review and editing, D.A.R.I. and G.N.H.A.; visualization, D.A.R.I. and G.N.H.A.; supervision, D.A.R.I. and G.N.H.A.; project administration, D.A.R.I. and G.N.H.A. All authors have read and agreed to the published version of the manuscript.

**Funding:** This research received no external funding.

**Institutional Review Board Statement:** Not applicable.

**Informed Consent Statement:** The study did not involve humans and animals.

**Data Availability Statement:** Not applicable.

**Acknowledgments:** We thank the General Hospital of Mexicali, Hiram Jaramillo, Ruiz Lujan, the microbiology department team, and our fellow intern colleagues who facilitated the realization of this article.

**Conflicts of Interest:** The authors declare no conflict of interest.

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
