# Peer review of "Bacterial Co-Infection in Patients with Coronavirus: A Rapid Review to Support COVID-19 Antimicrobial Prescription"

_2036-7481, doi:10.3390/microbiolres14040111_

Round 1

Reviewer 1 Report

Dear Authors,

Your manuscript entitled "Bacterial co-infection in patients with coronavirus: A rapid review to support COVID-19 antimicrobial prescription" was reviewed.

This paper deserves attention since it highlights an important topic related to detection of the co-infection in COVID-19 patients, which is considered as a major public health problem during pandemic (example SARS-CoV-2 one).

The article is well presented, its design is good, but it need several modifications, regarding the English language, the title, the Figures, etc.

Kindly find below a list of my minor and major comments:

Minor Comments:

01- In some place in the manuscript you talk about this paper as a review (example: line 54) but it seems as an original research article, Authors are invited to remove the term review and replace it by the term "article" or "study"

02- In the Abstract section, Line 1, Authors are invited to replace "in patients positive for COVID-19" by "in COVID-19 patients".

03- In the Abstract section, Line 6, Authors are invited to add "for COVID-19" after the sentence "The 1663 patients patients had positive results".

04- In the Whole manuscript, Authors are invited to put the full term followed by its abbreviation between parenthesis, examples are the follow:

a) Coronavirus Disease 2019 (COVID-19).

b) Severe Acutes Respiratory Syndrome Coronavirus 2 (SARS-CoV-2).

c) World Health Organization (WHO).

etc.

05- In the Whole manuscript, Authors are invited to put the full name of bacteria followed by the abr between parenthesis, example: 

a) Acinetobacter baumannii (A. baumannii).

a) Pseudomonas aeruginosa (P. aeruginosa).

a) Staphylococcus aureus (S. aureus).

d) etc.

06- In the Introduction section, After the Line 21, Authors are invited to talk about the following point: Several risks play an important role in the occurrence of COVID-19 in humans, such as variant of concern, blood group, place of residence, etc. And several conditions can increase the severity of the disease such as the age of patients, the immune system of the host, and the presence of a co-infection. For these points authors can add the following references in addition to athors:

Ref 1: Risk Markers of COVID-19, a Study from South-Lebanon

Ref 2: The emergence of SARS-CoV-2 variant (s) and its impact on the prevalence of COVID-19 cases in the Nabatieh Region, Lebanon

07- In the Introduction section, Line 22, Authors are invited to replace "such as viruses" by "such as other viruses".

08- In the Introduction section, Line 32, Authors are invited to correct the term "co-infection".

09- In the Introduction section, Authors are invited to move the reference "10" from the line 72 to the line 70.

10- Since Authors talked about candida in the manuscript, so they are invited to change the title of this paper since it not limited to bacterial co-infections.

11- In the Discussion section, Line 124, The expression "At this point" is not clear.

12- In the Conclusion section, Line 187, Authors are invited to replace "the spectrum of microorganisms causing COVID-19" by "the spectrum of microorganisms causing co-infections with SARS-CoV-2".

Major Comments:

01- In the Materials and Methods section, In several points authors are presenting results in this section, or it is not acceptable, Authors are invited to correct this mistake.

02- In the Results section, The First paragraph of this section and the legends of the Figure 1 are the same. Authors are invited to correct this point.

03- In the Results section, Authors are invited to put another figure because the present one is not clear and not informative for readers.

Best Regards,

Dear Authors,

The paper needs moderate English language polishing,

You can refer to an English native speaker for this.

Best Regards,

Reviewer 2 Report

The authors of this study conducted an analysis of microorganisms in patients co-infected with SARS-CoV-2 at a single center hospital. However, several key pieces of information are missing from this study, which hinders a clear interpretation of the results based solely on the ranking of microorganisms in the reported co-infection patients.

Firstly, the study does not provide information about whether the patients included in the analysis were in-patients, out-patients, or receiving intensive care. This information is essential for understanding the severity and clinical context of the co-infections.

Secondly, it is not clear whether the infections observed in the patients were community-acquired or hospital-acquired. Different sources of infection can have distinct implications for transmission dynamics, treatment strategies, and infection control measures.

Additionally, the study does not address the incidence of drug resistance among the identified microorganisms. Assessing the presence and extent of drug resistance is critical for guiding appropriate treatment choices and understanding potential challenges in managing these co-infections.

These missing pieces of information are crucial for accurately interpreting the findings and drawing meaningful conclusions from the analysis. Without them, it is difficult to provide comprehensive recommendations or draw definitive conclusions. Therefore, it is necessary to consider these critical factors in future studies to ensure a more thorough understanding of co-infections in SARS-CoV-2 patients.

I could understand most of the writing but could be improved.

Reviewer 3 Report

The authors satisfactorily analyzed the literature on the problem, but missed such a significant cause of coinfection with pathogenic microflora in patients with COVID-19 as the development of pneumonia or a complication of the course of lung damage during mechanical ventilation (doi: 10.1371). /journal.pone.0246318; doi: 10.3390/life11070601), especially in severe illnesses (10.12998/wjcc.v9.i31.9481). Mechanical ventilation has been suggested to cause death or severe post-COVID syndrome (doi: 10.3390/jcm11010224.).

In preparing the review, the authors did not use extensive literature on the problem, which reduced the information content and quality of the work (doi: 10.3390/life11070601; .doi: 10.3390/microorganisms908177; doi: 10.1007/s10096-020-04063-8), etc.

Conclusion: the work needs significant adjustment. It is advisable to consider a wider range of publications on the problem, to highlight the effect of mechanical ventilation on co-infection of patients with COVID-19 with pathogenic and opportunistic microflora.

Round 2

Reviewer 1 Report

Dear Authors,

Thank you for the modifications you did,

I have some modifications that need to be answered, such the follow:

In the Introduction section, After the Line 21, Authors are invited to talk about the following point: Several risks play an important role in the occurrence of COVID-19 in humans, such as variant of concern, blood group, place of residence, etc. And several conditions can increase the severity of the disease such as the age of patients, the immune system of the host, and the presence of a co-infection. For these points authors can add the following references in addition to authors:

Ref 1: Risk Markers of COVID-19, a Study from South-Lebanon

Ref 2: The emergence of SARS-CoV-2 variant (s) and its impact on the prevalence of COVID-19 cases in the Nabatieh Region, Lebanon.

Best Regards,

Some English sentences need paraphrasing.

Author Response

Thank you very much for your comments and suggestions. The references suggested were read and added to the document. Complement the ideas of the introduction.

Reviewer 2 Report

After the revision, the manuscript has undergone significant improvements, making it much easier to interpret the results. While I appreciate the enhancements, I couldn't find specific information in the manuscript about whether all samples subjected to PCR analysis also underwent culture testing. To gain clarity on this matter, I recommend carefully revisiting the text, especially the methodology or materials and methods section, which should provide details on the sample collection and testing procedures.

No apparent revision needed.

Author Response

We appreciate your opinion and suggestions. This study was obtained from the database provided by the General Hospital of Mexicali, fulfilling two inclusion criteria in this study. Patients admitted with suspicion of Covid-19 and favorable to the confirmatory RT-PCR test. Nasopharyngeal exudates were collected from these patients and cultured to isolate in case of any microbial growth. The MALDI-TOF technique identified the colonies obtained. It is a work generated from the information obtained from the files of the Hospital described in the study period from 2020 to 2021, which was the most critical stage of the pandemic in the city of Mexicali.

Reviewer 3 Report

Unfortunately, the authors limited themselves to editorial corrections, which clearly did not improve the content of the article. The authors did not consider it necessary to take into account the recommendations for improving the review of the literature in terms of the sources cited. I don't see an option to approve an article for publication.

Author Response

We have read the suggestions provided and the references cited. Your opinions are highly valued. This work aims to identify the most prevalent bacteria in patients positive for Covid-19 admitted to the General Hospital of Mexicali. In Mexico, there is the experience of the 2009 influenza pandemic, in which the bacteria that caused co-infections were identified. Identifying these bacteria during the SARS-CoV2 pandemic with the current problem of increased bacterial resistance is opportune. This information is relevant to the local medical community and generates relevant knowledge in the face of a new Covid-19 outbreak. Understanding the causes of co-infections is not the objective of this publication. At the same time, we are analyzing the resistance profile of the identified microorganisms that will generate future publications. It is the main reason that describing the causes of co-infection in this work, rather than strengthening it, could generate isolated ideas since we need all the information in this regard.

Round 3

Reviewer 1 Report

Dear Authors,

Thank you for the modifications you did,

The article is better for publication in its present form,

Best Regards,

Author Response

Thank you.

We appreciate your suggestions. 

Regards

Reviewer 2 Report

No further comments.

Author Response

(The authors gave the same response as above.)

Reviewer 3 Report

The authors made some editorial improvements, which, however, did not conceptually change the previous version of the manuscript. As before, the risk of transferring a patient to artificial lung ventilation on the likelihood of developing coinfections is completely insufficiently covered. The final version of the manuscript, in my opinion, does not quite correspond to the level of publication in the journal. I think the article should be rejected.

Author Response

We appreciate your suggestion. 

The risk of transferring a patient to artificial lung ventilation on the likelihood of developing coinfections is insufficiently covered.

This concept is not objective to this article. We identified the microorganisms with more prevalence of co-infection; that information has not been evaluated previously.

Regards

Round 4

Reviewer 3 Report

My opinion on this article has not changed. I still feel that publication of the manuscript is more appropriate in a specialized journal such as COVID (ISSN 2673-8112).

Author Response

In the last paragraph of the discussion of lines 180-205, we added an analysis of the suggestion. Another four references were added to this topic, and they are Ref23, Ref24, Ref25, and Ref26. They are updated references and suggested reading by Reviewer 3.

Regards